# Antineoplastic kinase inhibitors: A new class of potent anti-amoebic compounds

**Conall Sauvey** [1] *, **Gretchen Ehrenkaufer** [2], **Da Shi** [1], **Anjan Debnath** [1],
**Ruben Abagyan** [1] *

**1** Center for Discovery and Innovation in Parasitic Diseases, Skaggs School for Pharmacy and Pharmaceutical Sciences, University of California—San Diego, La Jolla, California, United States of America, **2** Division of Infectious Diseases, Department of Internal Medicine, Stanford University School of Medicine, Stanford, California, United States of America

* csauvey@gmail.com (CS); rabagyan@health.ucsd.edu (RA)

## Abstract

*Entamoeba histolytica* is a protozoan parasite which infects approximately 50 million people worldwide, resulting in an estimated 70,000 deaths every year. Since the 1960s *E. histolytica* infection has been successfully treated with metronidazole. However, drawbacks to metronidazole therapy exist, including adverse effects, a long treatment course, and the need for an additional drug to prevent cyst-mediated transmission. *E. histolytica* possesses a kinome with approximately 300–400 members, some of which have been previously studied as potential targets for the development of amoebicidal drug candidates. However, while these efforts have uncovered novel potent inhibitors of *E. histolytica* kinases, none have resulted in approved drugs. In this study we took the alternative approach of testing a set of twelve previously FDA-approved antineoplastic kinase inhibitors against *E. histolytica* trophozoites *in vitro*. This resulted in the identification of dasatinib, bosutinib, and ibrutinib as amoebicidal agents at low-micromolar concentrations. Next, we utilized a recently developed computational tool to identify twelve additional drugs with human protein target profiles similar to the three initial hits. Testing of these additional twelve drugs led to the identification of ponatinib, neratinib, and olmutinib were identified as highly potent, with $EC_{50}$ values in the sub-micromolar range. All of these six drugs were found to kill *E. histolytica* trophozoites as rapidly as metronidazole. Furthermore, ibrutinib was found to kill the transmissible cyst stage of the model organism *E. invadens*. Ibrutinib thus possesses both amoebicidal and cysticidal properties, in contrast to all drugs used in the current therapeutic strategy. These findings together reveal antineoplastic kinase inhibitors as a highly promising class of potent drugs against this widespread and devastating disease.

## Author summary

Every year, nearly a hundred thousand people worldwide die from infection by the intestinal parasite *Entamoeba histolytica*, despite the widespread availability of metronidazole as a treatment. Here we report that six anticancer drugs of the kinase inhibitor class possess potent anti-amoebic properties, with one of them killing both actively dividing parasite

**Data Availability Statement:** All relevant data are within the manuscript and its Supporting Information files.

**Funding:** AD received funding from the United States National Institutes of Health (grant no. R21AI146460). CS received funding from the

University of California - San Diego Global Health Institute (http://globalhealth.ucsd.edu/grant-recipients/2017- 2018/Pages/GraduateStudentResearchers.aspx) (GHI Graduate Student research grant 2017-2018) GE received funding from the SPARK Translational Research Program at Stanford University (http://med.stanford.edu/sparkmed.html), the United States National Institute of Health (https://www.nih.gov/) (grant no. R21-AI123594), and the United States National Institute of Health National Center for Advancing Translational Science (https://ncats.nih.gov/) (grant no. UL1-TR001085) DS and AD received funding from the United States National Institute of Health (https://www.nih.gov/) (grant no. 1KL2TR001444) RA received funding from the United States National Institute of Health National Institute of General Medical Sciences (https://www.nigms.nih.gov/) (grant no. GM071872) The funders had no role in study design, data collection and analysis, decision to publish, or preparation of the manuscript.

**Competing interests:** The authors have declared that no competing interests exist.

and its transmissible cysts. These anticancer kinase inhibitors, including the dual-purpose drug with both amoebicidal and cysticidal activities may be used to treat amoebiasis, especially in cancer patients or in life-threatening brain- and liver-infecting forms of the disease.

## Introduction

*Entamoeba histolytica* is a parasitic amoeba which infects an estimated 50 million people worldwide, resulting in around 70,000 deaths per year [1]. *E. histolytica* infection is known as amoebiasis and primarily affects the intestinal tract in humans, most commonly causing symptoms such as abdominal pain, bloody diarrhea, and colitis [2]. In rare cases the infection spreads to other organs such as the liver and brain, and in serious cases results in patient death [2]. *E. histolytica*'s life cycle consists of a trophozoite vegetative stage which matures in its host to an infective cyst stage. The cyst stage is excreted in the host's feces, infecting a new host when ingested via a route such as drinking contaminated water. In the majority of cases where *E. histolytica* is ingested it lives asymptomatically in the human host's intestinal tract. Symptoms can develop when compromise of the mucosal layer allows it to come into contact with the intestinal wall, at which point it invades the wall and surrounding tissue causing characteristic 'flask-shaped ulcers' [3]. Due to this mode of transmission *E. histolytica* disproportionately affects populations experiencing sanitation problems associated with low socioeconomic status [2,4,5]. Malnutrition is also known to be a major risk factor for amoebiasis, especially in children [6].

*E. histolytica* infection is currently treated with the 5-nitroimidazole drug metronidazole, which has been in use since the 1960s and has widespread use as a treatment against anaerobic microbial infection [7,8]. However, while successful, metronidazole is not a perfect solution to *E. histolytica* infection, with a few particularly notable existing issues. One of these is problems with lack of patient compliance with the full course of treatment, leading to relapses and increased disease spread [7]. This is possibly due to factors such as drug adverse effects or the need for continued dosing past the resolution of disease symptoms [9,10]. Another issue is metronidazole's inability to kill the infective cyst stage of *E. histolytica*. Because of this, along with its complete absorbance from the intestines, metronidazole must be followed by a secondary luminal amoebicide such as paromomycin to prevent spread of the disease [11,12]. Also concerning is the potential for the emergence of resistance to metronidazole, which has been previously observed in the laboratory [13]. When considered together, these factors comprise an unmet need for alternative amoebiasis therapies.

Several efforts to find such alternative therapies have been undertaken over the years, including the recent development of the antirheumatic drug auranofin as a promising potential treatment for amoebiasis [14–16]. One noteworthy direction of anti-amoebic drug research has been the efforts of multiple groups to inhibit *E. histolytica* by targeting specific kinase proteins believed to be critical to the parasite's functioning [17–19]. This approach has often involved computational modeling and *in-silico* screening of compounds against the kinases of interest, followed by *in-vitro* tests of top-scoring molecules [17]. These efforts have resulted in both the discovery of potent new hit compounds as well as validation of the previously discovered activity of auranofin [17]. However, despite these successes, no new clinical treatments have yet been produced.

Importantly, one promising area currently unexplored by such studies is the potential of existing human kinase inhibitor drugs. A particular advantage of these drugs is the rich array

of data available regarding their activity profiles against human target proteins, which allows for the mapping and utilization of their complex multi-target pharmacology. Such maps could in turn be projected into the *E. histolytica* proteome and used to infer potential antiamoebic drug activity by identifying drugs with similar target profiles to known active compounds. We have previously published a computational tool capable of such mapping for antineoplastic drugs, including a large number of kinase inhibitors [20]. We describe here the use of this tool to prioritize molecules for screening against *E. histolytica* trophozoites based on initial hits from a small primary screen. In total, 6 antineoplastic kinase inhibitors (AKIs) were found to have potent and rapid anti-amoebic activity. The results of these experiments demonstrate the promise of using target-based analysis to leverage compounds with multi-target pharmacology against a human parasite.

## Materials and methods

### *E. histolytica* cell culture

*E. histolytica* strain HM-1:IMSS trophozoites were maintained in 50ml culture flasks (Greiner Bio-One) containing TYI-S-33 media, 10% heat-inactivated adult bovine serum (Sigma), 1% MEM Vitamin Solution (Gibco), supplemented with penicillin (100 U/mL) and streptomycin (100 μg/mL) (Omega Scientific) [14].

### Compounds

Compounds for screens were purchased from Fisher Scientific and Millipore-Sigma.

### Cell viability screen to determine drug potency against *E. histolytica*

Following a previously-published approach [14] *E. histolytica* trophozoites maintained in the logarithmic phase of growth were seeded into 96-well plates (Greiner Bio-One) at 5,000 cells/well to a total volume of 100 μl/well. 8- or 16-point two-fold dilution series of the treatment compounds were prepared, beginning at a maximum final treatment concentration of 50 μM. 0.5 μl of each drug concentration was added to triplicate wells for each treatment group. 0.5 μl of DMSO was used as a negative control, and 0.5 μl of 10 mM metronidazole dissolved in DMSO was used as a positive control, giving a final concentration of 50 μM. Alternatively, wells with only media were used as a negative control. The plates were placed in GasPak EZ (Becton-Dickinson) bags and incubated at 37˚C for 48hr. Plates were removed and 50 μl of CellTiter-Glo (Promega) was added to each well. Plates were shaken and incubated in darkness for 20 minutes and the luminescence value of each well was read by a luminometer (EnVision, PerkinElmer). Percent inhibition was calculated by subtracting the luminescence values of each experimental data point from the average minimum signal obtained from positive control values and dividing by the difference between the average maximum signal negative control and the positive control. The resulting decimal value was then multiplied by 100 to give a percentage.

### Determination of drug EC$_{50}$ values *in vitro* over time

Effects of different concentrations of compounds on *E. histolytica* trophozoite cell viability were determined as described in the previous section at a series of timepoints ranging from 6 hours to 48 hours following drug administration. EC$_{50}$ values were calculated at each time-point as previously described.

## Determination of varying drug exposure time effects

*E. histolytica* trophozoites were treated with either 5μM ponatinib, 5μM neratinib, 5μM olmutinib, or 10μM metronidazole in replicates of 4 wells in 96-well plates. At timepoints ranging from 2 to 48 hours, wells were aspirated, washed once with fresh media, and refilled with fresh media. At 48 hours, percentage trophozoite inhibition was measured using luminescence and calculated as described previously for each timepoint.

## Identification of desired target profile of active drugs

A desired target profile of active drugs was generated using the "Multi-drug target finder" tool in the CancerDrugMap (http://ruben.ucsd.edu/dnet/maps/drug_find.html) [20]. Drugs that were active (dasatinib, bosutinib, ibrutinib. . .) and inactive (nilotinib, imatinib. . .) in the *E. histolytica* proliferation assay were inputs, respectively. Drug-target interaction activity data for the tool were collected from multiple sources as previously described, including ChEMBL, PubChem, and literature sources [20]. Drug targets were ranked based on the drug-target activity data and using the following equations and assembled into the final anti-amoebic activity-associated profile.

Where: *Score of target S* $= \sum_{drugs}$ *weight* $\times$ *(pAct−4)*

$$pAct = -log(IC50/Kd/Ki)$$

For active drugs: *weight* = 1

For inactive drugs: $weight = -0.6 \times \sqrt{\frac{number\ of\ active\ drugs}{number\ of\ inactive\ drugs}}$

## Identification of drugs with desired target profile

A list of drugs with target profiles matching the desired target profile was generated with the "Multi-target drug finder" tool in CancerDrugMap (http://ruben.ucsd.edu/dnet/maps/tar_find.html). The desired target profiles generated above were input correspondingly. Resulting cancer drugs were ranked based on the drug-target activity and the following equations.

$$Score\ of\ drug\ S = \sum_{target} weight \times (pAct - 4)$$

Where

$$pAct = -log(IC50/Kd/Ki)$$

For targets to hit: *weight* = 1

For targets to avoid: $weight = -0.6 \times \sqrt{\frac{number\ of\ wanted\ targets}{number\ of\ unwanted\ targets}}$

Top-ranking drugs were then selected for further testing.

## Identification of potential targets of active drugs in the *E. histolytica* genome

15 top ranking human protein targets (YES1, ABL1, BTK, BMX, LCK, HCK, FGR, BLK, ERBB4, LYN, FYN, SRC, CSK, ABL2, and FRK) from the identified desired target profile were searched against the *E. histolytica* genome downloaded from (https://amoebadb.org/common/downloads/Current_Release/EhistolyticaHM1IMSS/). Human protein sequences were downloaded from (uniprot.org) and the annotated kinase domains of each protein were compiled into a single file. Full gapped optimal sequence alignments with zero end gap penalties

(ZEGA) were performed between the kinase domain sequences of the 15 targets and the *E. histolytica* genome. The significance of each alignment was assessed according to a number of residue substitution matrices as a pP value (pP = -log(P-value)) [21].

*E. histolytica* genes with pP over 10, namely the P-value of the alignment lower than $10^{-10}$ were selected as potential targets. A network map of the 15 top ranking human genes and homologous *E. histolytica* genes was generated with Graphviz neato, with edges corresponding to the pP values between human and *E. histolytica* genes.

### Cyst killing assay

For assays on mature cysts, a transgenic *E. invadens* line stably expressing luciferase (CK-luc) was used [22]. Mature cyst viability assay was performed as described previously [16]. Parasites were induced to encyst by incubation in encystation media (47% LG) [23]. After 72 h, parasites were washed once in distilled water and incubated at 25°C for 4–5 h in water to lyse trophozoites. Purified cysts were pelleted, counted to ensure equal cyst numbers, and resuspended in encystation media at a concentration of 1-5x10$^5$ cells per ml. One ml suspension per replicate was transferred to glass tubes containing encystation media and drug or DMSO, then incubated at 25°C for 72 h. On the day of the assay, cysts were pelleted and treated once more with distilled water for 5 h to lyse any trophozoites that had emerged during treatment. Purified cysts were then resuspended in 75 µl Cell Lysis buffer (Promega) and sonicated for 2x10 seconds to break the cyst wall. Luciferase assay was performed using the Promega luciferase assay kit according to the manufacturer's instructions. Assays were performed on equal volume of lysate (35 µl) and not normalized to protein content. Effect of the drug was calculated by comparison to DMSO control, after subtraction of background signal. Significance of drug effects was calculated using a one-tailed T-test.

## Results

### Screen of antineoplastic kinase inhibitors against *E. histolytica* trophozoites

In order to identify anti-amoebic activity among antineoplastic kinase inhibitors, a selection of 12 drugs was screened against *E. histolytica* trophozoites *in vitro*. All compounds tested were FDA-approved cancer chemotherapy drugs designed to inhibit human kinase proteins as their mechanism of action. *E. histolytica* trophozoites were seeded into 96-well plates along with a serially-diluted range of drug concentrations. Trophozoites were incubated for 48 hours, after which the surviving cell amount was determined using a luciferase-based cell viability assay. Percent inhibition of trophozoite growth was calculated for each treatment well in comparison with vehicle-only negative controls representing 0% inhibition, and media-only or metronidazole-treated positive controls representing 100% inhibition. From this data EC$_{50}$ values were calculated for each respective drug (Table 1). Out of the 12 drugs tested, ibrutinib, dasatinib, and bosutinib all were found to possess EC$_{50}$ values similar to or lower than the EC$_{50}$ values of of 2–5 µM for the currently used drug metronidazole (Fig 1). Based on these results we concluded that antineoplastic kinase inhibitor drugs are capable of potent inhibition of *E. histolytica* and determined that further analysis and refinement was warranted in order to discover even more potent drugs in the same class.

### Analysis of hits

In order to identify potent anti-amoebic candidates from the existing pool of AKI drugs, an approach was utilized where human protein target profiles were computationally generated for drugs active in the screen, followed by identification of additional drugs with matching or

**Table 1. Results of primary screen of AKI drugs against *E. histolytica* trophozoites.** Color scale indicates drug potency: darker blue = more potent, lighter blue = less potent. Anti-amoebic activity classified based on $EC_{50}$ value as: Very high (0.001–0.999 µM), High (1.000–4.999 µM), Moderate (5.000–9.999 µM), Low (10.000–19.999 µM), Very low (20.000–99.999 µM), or None ($EC_{50}$ > 100.000 µM).

| Drug name | $EC_{50}$ (µM) | Anti-amoebic activity |
|---|---|---|
| Ibrutinib | 0.98 | Very high |
| Dasatinib | 1.57 | High |
| Bosutinib | 1.94 | High |
| Nilotinib | 8.22 | Moderate |
| Gefitinib | 8.52 | Moderate |
| Sunitinib | 10.09 | Low |
| Afatinib | 11.69 | Low |
| Crizotinib | 12.83 | Low |
| Erlotinib | 42.07 | Very low |
| Dabrafenib | 55.18 | Very low |
| Vemurafenib | >100 | None |
| Imatinib | >100 | None |

similar target profiles. To generate the target profiles for active drugs, a computational tool called CancerDrugMap (CDM) was utilized, which we have previously described, and which is available at: *ruben.ucsd.edu/dnet/* [20]. Using CDM, human targets of both the active and inactive compounds from the initial were compared. Protein targets were scored and ranked based on targeting activity of active compounds as well as lack of targeting activity by the inactive ones. This generated a profile of human protein targets associated with the active amoebicidal drugs in the screen (Fig 2). Based on this profile, two strategies were then employed to identify drugs with similar target profiles and hence the potential for similar anti-amoebic activity (Fig 3). In the first strategy, CDM was used to score and rank all AKI drugs in the database based on their activity against the complete ranked target profile, using an algorithm described in Materials and Methods. Drugs possessing a score greater than or equal to 15 were selected for

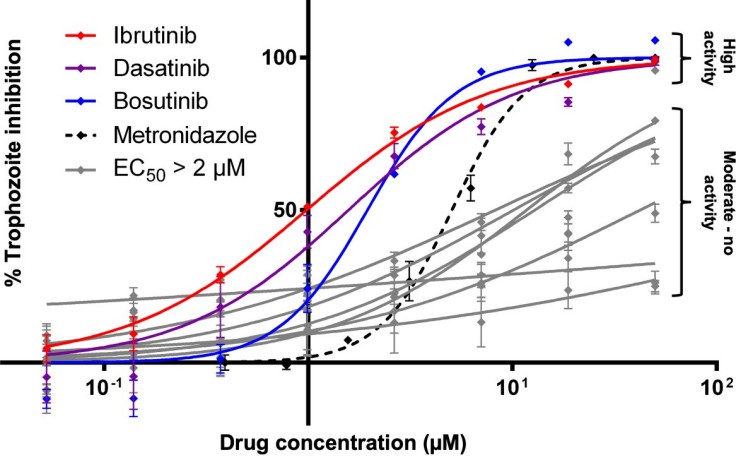

**Fig 1. $EC_{50}$ determination of antineoplastic kinase inhibitors against *E. histolytica* trophozoites.** Dose response curve plotting percentage inhibition of *E. histolytica* trophozoites compared to drug concentrations of antineoplastic kinase inhibitors. Trophozoites were assayed for cell viability following treatment with each drug for a period of 48 hours. The three drugs with the lowest $EC_{50}$ values (ibrutinib, dasatinib, and bosutinib) are plotted in red, purple, and blue. All drugs with $EC_{50}$ values > 2 µM are plotted in gray. Each data point represents mean values of percentage inhibition. Error bars represent standard deviation. Complete list of drugs can be found in Table 1.

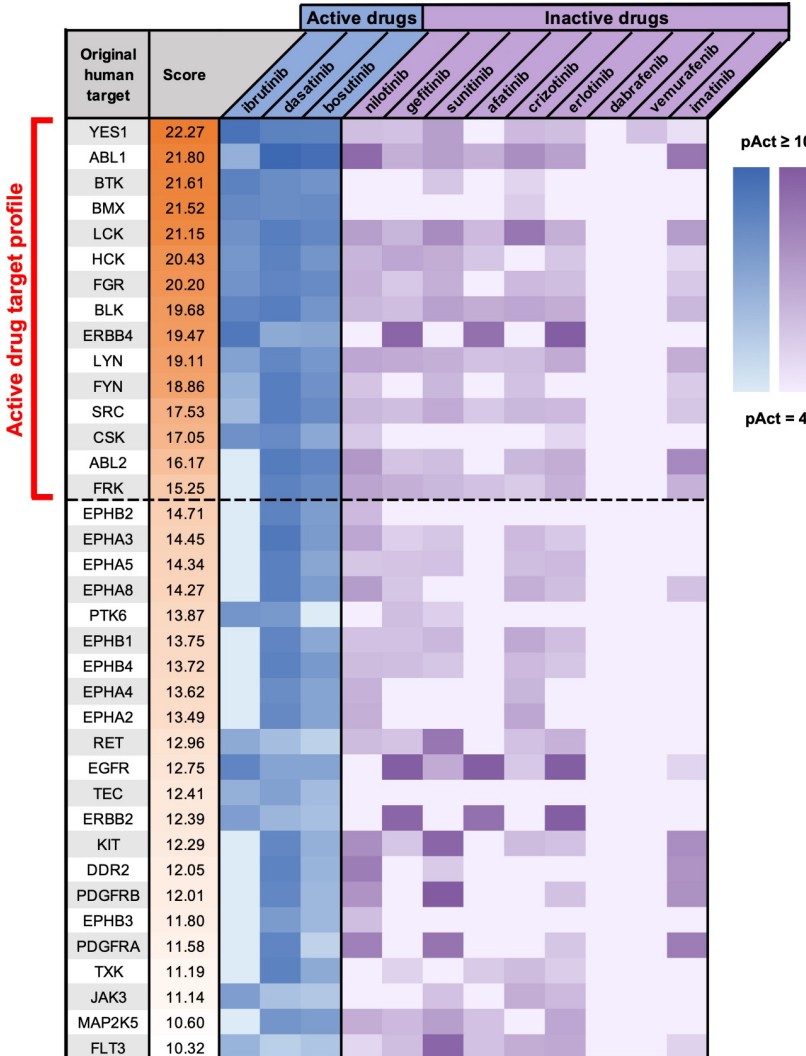

**Fig 2. Generation of human target profile for inhibitors of *E. histolytica*.** Human kinase proteins scored and ranked based on targeting activity data for active drugs versus inactive drugs from the screen. Heatmap represents the calculated activity values (pAct, see Materials and Methods) of individual drugs against individual human protein targets. Darker colors indicate stronger drug activity against the protein. Dashed line represents the cutoff pAct value for proteins to be included in the target profile for the purpose of identifying additional *E. histolytica* drug candidates.

further *in vitro* testing (Fig 4). This cutoff value was chosen based on a naturally occurring gap between drugs with a score of 15 and the next-lowest scoring drugs, with the latter possessing only few and very weak activity scores against any of the target profile proteins. The second strategy was identical to the first with the exception that CDM was used to score and rank drugs based on activity against target proteins individually rather than the complete profile. Drugs active above a threshold score of 15 against individual proteins from the target profile were ranked by the number of proteins from the profile they possessed this level of activity against (Table 2). Drugs possessing a score at or over the threshold of 15 against more than 2 proteins from the target profile were selected for further investigation. The first and second strategies combined generated a list of 15 drugs with similar known human protein targets to the positive hits from the initial screen, and hence high potential for corresponding anti-amoebic activity.

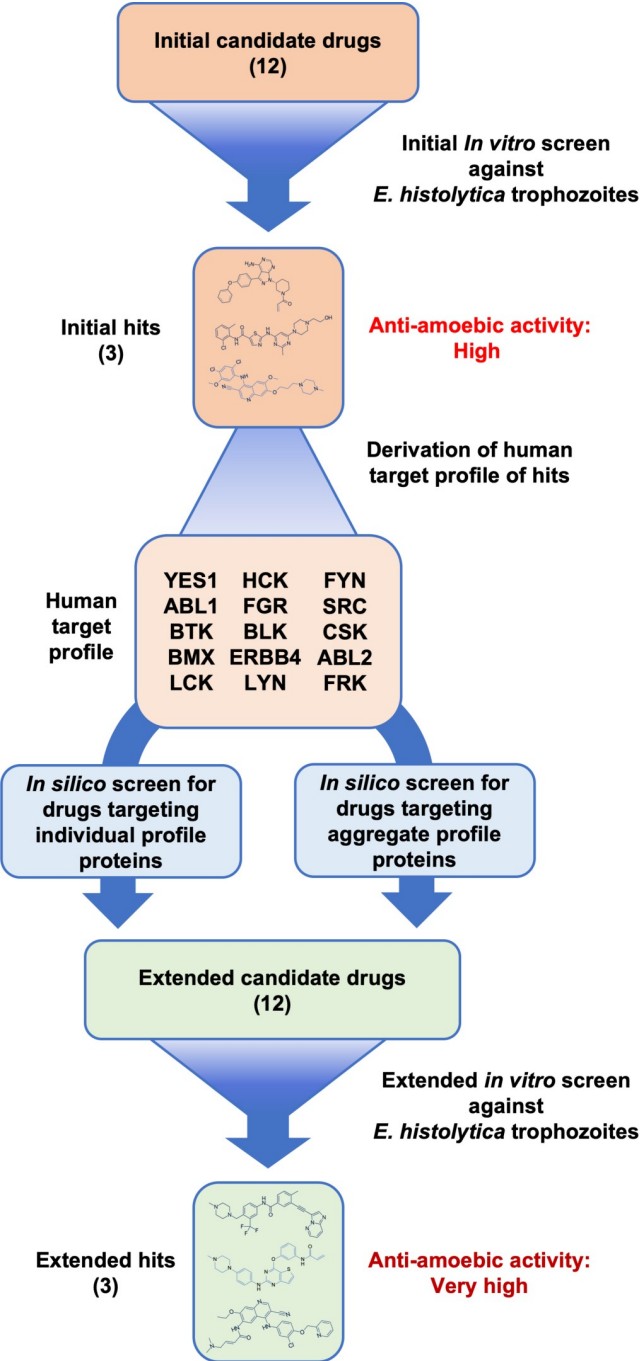

**Fig 3. Graphical screening workflow of antineoplastic kinase inhibitors against *E. histolytica*.** Chemical structures represent the three drugs found to possess the lowest EC$_{50}$ values in each screen (ibrutinib, dasatinib, bosutinib, and ponatinib, neratinib, olmutinib respectively).

## Potential drug target proteins are present in the *E. histolytica* proteome

While the data regarding activity of AKI drugs against human target proteins is valuable for the purpose of grouping drugs with the potential for similar activity against *E. histolytica*, it does not provide information regarding the drugs' actual protein targets in the parasite. In

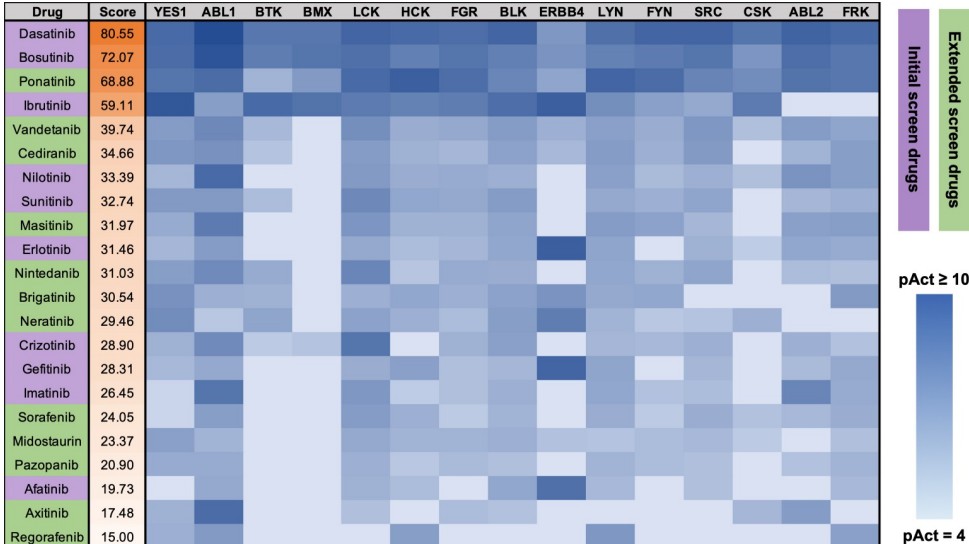

**Fig 4. In silico screen based on human target profile to determine new potent amoebicidal drug candidates.**
Antineoplastic kinase inhibitor drugs scored and ranked based on activity data regarding all 15 proteins in the amoebicidal drug target profile shown in Fig 3. Score shown in the second column is calculated from the weighted sum of pAct values (see Materials and Methods.) Shown are all drugs meeting the cutoff score of 15 for further screening. Heatmap displays the calculated pAct of individual drugs against individual protein targets. Darker colors indicate stronger drug activity against the protein. Purple highlight indicates drugs included in the initial *in vitro* screen. Green highlight indicates new candidate drugs.

order to identify whether potential drug targets exist in *E. histolytica* that are similar to the human target profile proteins, the human proteins were searched against the *E. histolytica* proteome. In order to do so, the kinase domain sequences of the human proteins were extracted and aligned against the complete published set of *E. histolytica* open reading frames (ORFs). 32 *E. histolytica* ORFs were found to align to the human sequences with a p-value of $10^{-10}$ or less. A network map was generated of these top-scoring *E. histolytica* proteins and their relationship to the human protein targets (Fig 5). In the network map multiple *E. histolytica* ORFs can be seen to possess strong alignments to several human sequences. These results demonstrate the possibility that *E. histolytica* may possess protein targets equivalent to those known to be targeted in humans by the active AKI drugs.

## Extended screen of candidate drugs based on primary analysis

Based on our CDM analysis we tested the list of 12 potentially active AKI drugs against *E. histolytica* trophozoites in an extended *in vitro* screen. Compounds were tested as previously, using the same luciferase-based cell viability assay to determine $EC_{50}$ values. The drugs ponatinib, neratinib, and olmutinib were found to possess highly potent activity in this screen, all with sub-micromolar $EC_{50}$ values (Table 3) (Fig 6).

## Hit compounds kill *E. histolytica* trophozoites

An important question regarding the activity of any compound intended to act against *E. histolytica* is whether it induces cell death in the parasite or merely slows its replication. In order to determine which type of activity belongs to each of the AKI drugs active in the initial and extended screens, we measured the number of surviving cells after 48 hours of drug treatment compared to freshly-counted aliquots of cells. 5,000 cells per well were seeded into 96-well

**Table 2. Analysis of drugs based on activity towards individual proteins in the active drug target profile.** Drugs are ranked based on the number of proteins from the active drug target profile towards which they possess an activity score greater than a threshold value. Darker color indicates a greater number of proteins. Drugs possessing the desired level of activity towards more than two target profile proteins were considered for further screening, shown above red line.

| Drug name | Number of target profile matches |
|---|---|
| Dasatinib | 15 |
| Bosutinib | 15 |
| Ponatinib | 13 |
| Ibrutinib | 11 |
| Vandetanib | 6 |
| Cediranib | 6 |
| Sunitinib | 4 |
| Nilotinib | 4 |
| Masitinib | 4 |
| Regorafenib | 3 |
| Nintedanib | 3 |
| Neratinib | 3 |
| Imatinib | 3 |
| Brigatinib | 3 |
| Acalabrutinib | 3 |
| Sorafenib | 2 |
| Olmutinib | 2 |
| Erlotinib | 2 |
| Crizotinib | 2 |
| Axitinib | 2 |
| Rociletinib | 1 |
| Osimertinib | 1 |
| Lapatinib | 1 |
| Gefitinib | 1 |
| Afatinib | 1 |

plates and treated with dasatinib, bosutinib, ibrutinib, ponatinib, neratinib, olmutinib, metronidazole, or vehicle. A concentration of 10µM was used for all drugs in order to ensure maximal and unambiguous inhibitory effects against the trophozoites. After 48 hours fresh aliquots containing a known number of cells were seeded into empty wells, CellTiter-Glo was added, and the luminescence of all wells was measured. Using the linear relationship of CellTiter-Glo luminescence to the number of cells being assayed, the number of cells in treatment group wells was calculated using their luminescence values relative to those of the freshly-aliquoted wells. All drugs tested were found to have significantly decreased the number of live cells in their treatment groups below the initial 5,000 cells. In contrast, cells treated with only vehicle significantly increased in number to over 14,000 cells per well (Fig 7). Interestingly, none of the drugs tested, including metronidazole, completely reduce the estimated cell number to zero. This could possibly be due to either a small number of genuine surviving cells, or to dying cells which still contain measurable ATP.

In order to characterize whether the active drugs genuinely kill *E. histolytica* trophozoites or merely act as false positives by inhibiting the ATP-driven, luciferase-based CellTiter-Glo assay system we tested concentration ranges of each drug on cells and immediately after

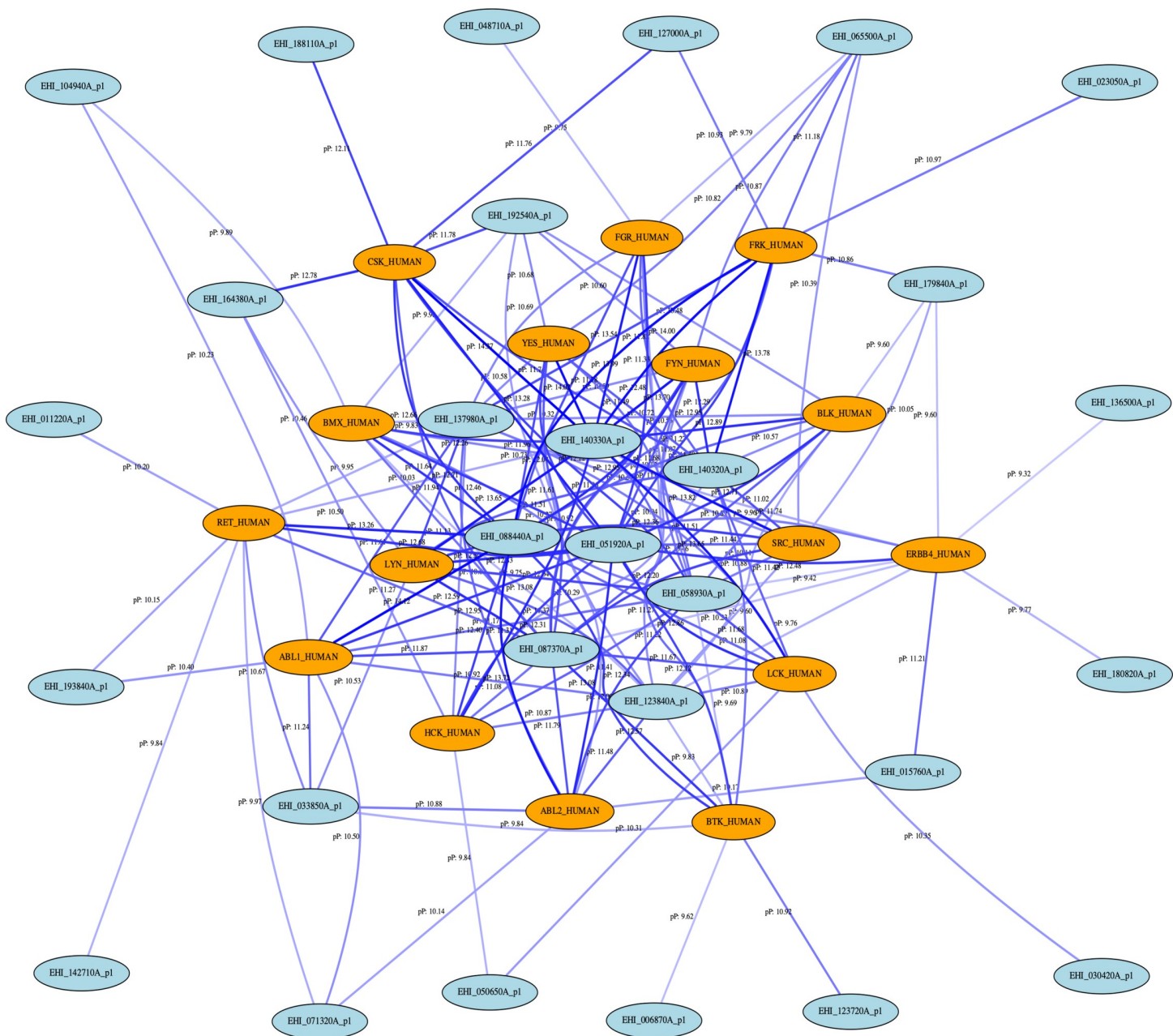

**Fig 5. Network map of active drug profile proteins with orthologous *E. histolytica* ORFs.** Orange ovals represent human sequences. Blue ovals represent *E. histolytica* sequences. Lines represent alignment relationships possessing a calculated pP greater than the cutoff value of 10. Line color represents pP value, with darker lines denoting higher pP. Blue ovals shown in the center with the highest number of connecting lines represent the most likely *E. histolytica* orthologs of the human target proteins.

addition of drugs. If the drugs were acting on the CellTiter-Glo assay reagents rather than the cells themselves, a dose-response relationship of drug to assay activity should have been evident. However, no dose-response relationship was observed, and measured luminescence values remained equivalent across all concentrations of drug treatments (S1 Fig). These results indicate that the active drugs are true positives against *E. histolytica* cells and do not inhibit the assay itself.

**Table 3. Results of extended screen of AKI drugs against *E. histolytica* trophozoites.** Color scale indicates drug potency: darker blue = more potent, lighter blue = less potent. Anti amoebic activity classified based on $EC_{50}$ value as: Very high (0.001–0.999 μM), High (1.000–4.999 μM), Moderate (5.000–9.999 μM), Low (10.000–19.999 μM), Very low (20.000–99.999 μM), or None ($EC_{50} > 100.000$ μM).

| Drug name | $EC_{50}$ (μM) | Anti-amoebic activity |
|---|---|---|
| Ponatinib | 0.1299 | Very high |
| Neratinib | 0.3113 | Very high |
| Olmutinib | 0.6462 | Very high |
| Nintedanib | 4.239 | High |
| Cediranib | 7.286 | Moderate |
| Vandetanib | 8.971 | Moderate |
| Acalabrutinib | 11.34 | Low |
| Masitinib | 14.03 | Low |
| Regorafenib | 15.94 | Low |
| Sorafenib | 18.3 | Low |
| Pazopanib | >100 | None |
| Axitinib | >100 | None |

## Hit compounds kill *E. histolytica* trophozoites as rapidly as metronidazole

In addition to drug potency, an important characteristic of any drug is the rapidity with which it achieves its desired effect. In order to characterize this aspect of the most active drugs from the trophozoite viability screens, the $EC_{50}$ values of ponatinib, neratinib, olmutinib, and metronidazole against *E. histolytica* trophozoites were measured at a series of timepoints after treatment initiation. The luciferase-based CellTiter-Glo cell viability assay was used as previously to determine the percent inhibition in each set of experimental replicates. Duplicate plates containing cells treated with serially-diluted ranges of ponatinib, neratinib, olmutinib, and metronidazole concentrations were prepared for each desired time point. Measurements were collected at 12, 24, 36, and 48 hours post-drug-treatment respectively. From the data

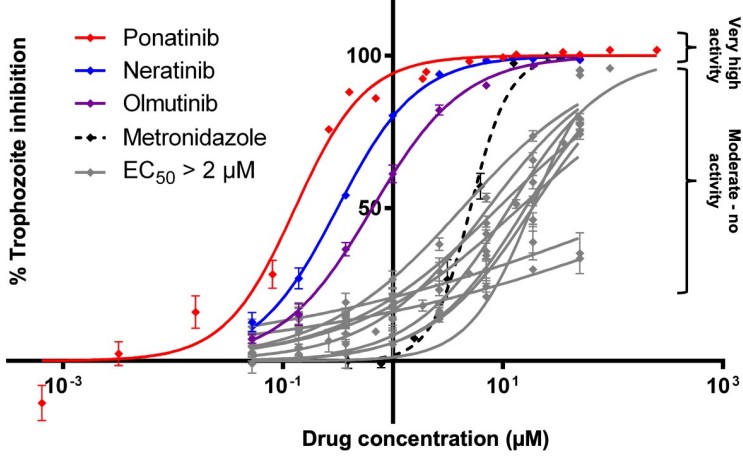

**Fig 6. Extended screen of antineoplastic kinase inhibitors against *E. histolytica* trophozoites.** Dose response curves plotting percentage inhibition of *E. histolytica* trophozoites at different drug concentrations. Trophozoites were assayed for cell viability following treatment with each drug for a period of 48 hours. The three drugs with the lowest $EC_{50}$ values (ponatinib, neratinib, and olmutinib) are plotted in red, purple, and blue. All drugs with $EC_{50}$ values > 2 μM are plotted in gray. Each data point represents mean values. Error bars represent standard deviation. Complete list of drugs tested found in Table 3.

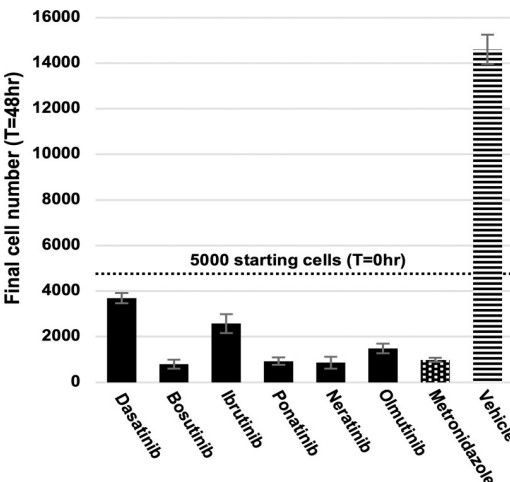

**Fig 7. Amoebicidal effects of antineoplastic kinase inhibitor drugs.** Live cell number calculated in comparison to aliquots of known amounts of cells. All drugs tested at 10μM, vehicle = 0.5% DMSO. Dotted line represents the 5000 cells originally seeded into all wells for each treatment group. Error bars represent standard deviation.

obtained $EC_{50}$ values were calculated for each time point of each drug treatment and compared over time (Fig 8A, 8B, 8C and 8D). All drugs tested achieved steady $EC_{50}$ values within 36 hours equivalent to those observed at 48 hours (Fig 8E). These results indicate that the active AKI drugs achieve their anti-amoebic effects as rapidly as the current treatment, metronidazole.

## Hit compounds and metronidazole require similar exposure times for *E. histolytica* trophozoite inhibition

To determine the amount of exposure time necessary for parasite killing by the compounds most active in the initial and extended screens, *E. histolytica* trophozoite inhibition was measured following varying treatment periods with either ponatinib, neratinib, olmutinib, or metronidazole. Cells were treated with drugs at 5 μM for intervals ranging from 2 to 48 hours, followed by drug washout and continued incubation to 48 total hours. This concentration was chosen to ensure complete parasite killing by all drugs after 48 hours. Trophozoite viability was measured for each drug and exposure time using CellTiter-glo and the percentage inhibition was calculated. Because each drug tested possess a different anti-amoebic $EC_{50}$ value, the percentage inhibition for each exposure time was scaled to the 48-hour value for each respective drug. As a result, the relative effectiveness of varying exposure times could be compared across drugs irrespective of varying drug potency. All drug treatments achieved inhibition levels after 24 hours of exposure time similar to levels observed after 48 hours (Fig 8F). These results indicate that ponatinib, neratinib, olmutinib, and metronidazole all require roughly 24 hours of parasite exposure time in order to achieve maximal levels of parasite inhibition. Additionally, both ponatinib and neratinib achieved significantly higher scaled % inhibition levels after 2, 6, and 12 hours compared to metronidazole, indicating that a somewhat shorter exposure time might be required for these drugs to achieve their antiparasitic effects.

## Antineoplastic kinase inhibitors kill mature *Entamoeba* cysts

A major drawback of metronidazole as a treatment for amebiasis is its poor activity against luminal parasites and cysts [2]. To determine if AKIs may be superior in this respect, we

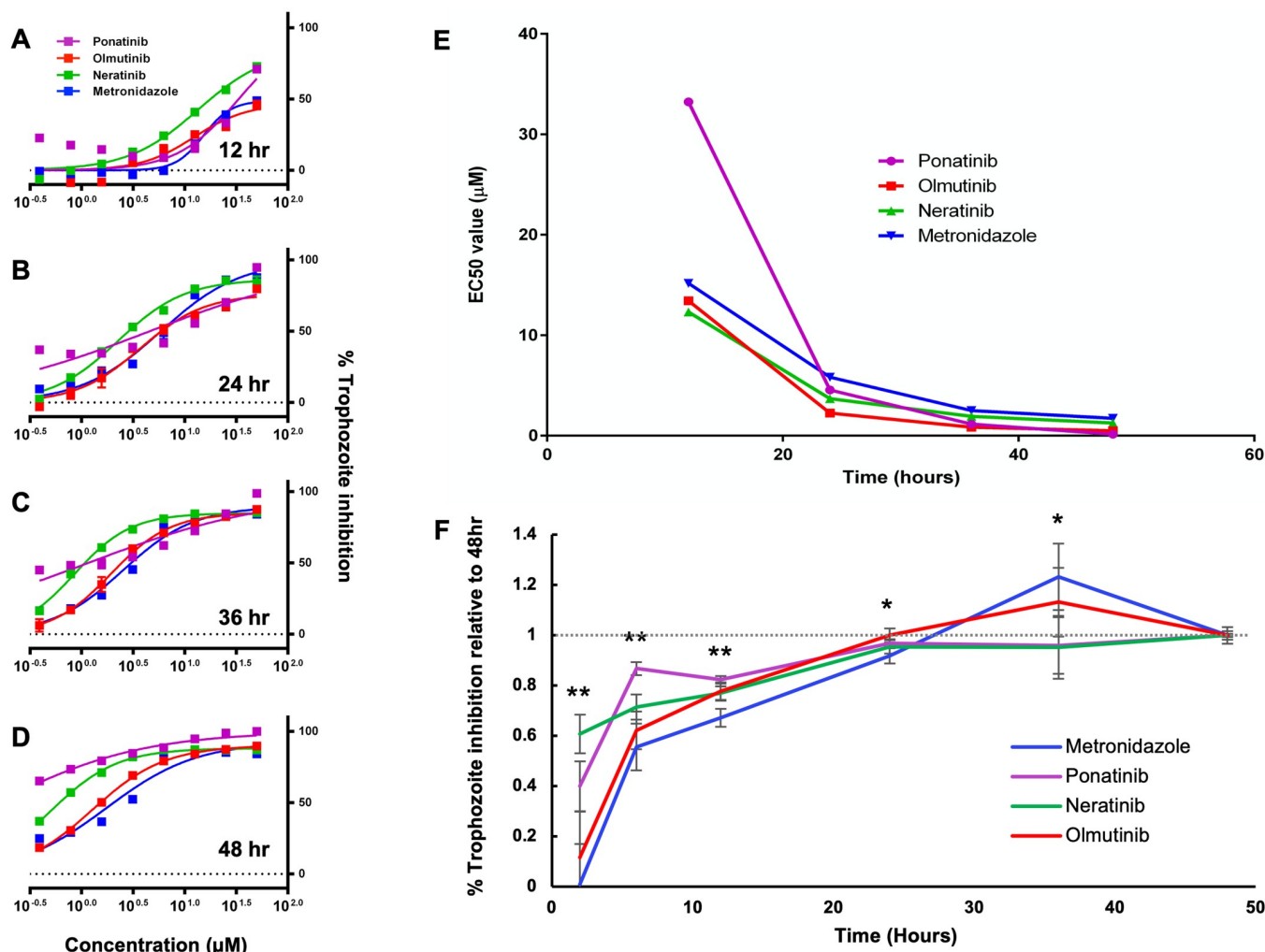

**Fig 8. Timing of drug action against *E. histolytica* trophozoites.** (A—D) Dose-response curves measured at 12, 24, 36, and 48hr for ponatinib, neratinib, olmutinib, and metronidazole. (E) Plot of $EC_{50}$ values calculated from the data shown in (A-D) graphed over time. (F) Plot of *E. histolytica* trophozoite inhibition by AKI drugs after varying exposure times. Trophozoites were treated with ponatinib, neratinib, olmutinib, or metronidazole at 5µM for 2, 6, 12, 24, 36, or 48hr, followed by drug washout and continued incubation until 48hr. Percent inhibition was then determined. Points represent % inhibition for each drug and exposure time scaled to the 48hr % inhibition value for the same drug. Statistical difference between groups was determined by 1-way ANOVA for each exposure time. (** = $p < 0.01$) (* = $p < 0.05$).

assayed for killing of mature *Entamoeba* cysts. As *E. histolytica* cannot be induced to encyst *in vitro* [23], the related parasite, *E. invadens*, a well-characterized model system for *Entamoeba* development, was utilized. Mature (72h) cysts of a transgenic line constitutively expressing luciferase were treated with 10 µM dasatinib, bosutinib, ibrutinib, or 0.5% DMSO as negative control, for 3 days. After treatment, cysts were treated with distilled water for five hours to remove any remaining trophozoites, and luciferase activity was assayed. Ibrutinib was found to significantly reduce luciferase signal to between 10% and 50% of controls, indicating that this AKI drug is capable of killing *Entamoeba* cysts. In contrast, metronidazole up to 20 µM had no effect (Fig 9). As ibrutinib is known to act as a covalent inhibitor of human kinase proteins, another such covalent inhibitor which showed activity in the extended screen, acalabrutinib, was also tested [24,25]. However, this drug did not consistently display any significant cysticidal activity.

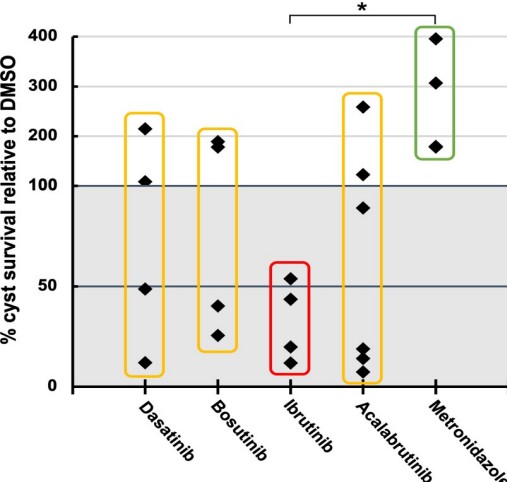

**Fig 9. Activity of antineoplastic kinase inhibitors against *Entamoeba* cysts.** Cyst survival measured using luminescence values of luciferase-expressing *E. invadens* cysts after drug treatments, compared with DMSO-treated controls. Cysticidal effect corresponds to cyst survival values below 100%. Metronidazole tested at 20μM. All other drugs tested at 10μM. Data points represent biological replicates. Asterisk indicates ($p < 0.05$).

## Discussion

Treatment options for amoebiasis are currently limited to either nitroimidazole drugs such as metronidazole, which acts via anaerobic activation to toxic reactive forms in *E. histolytica* [10]. While other drugs have been proposed or used at times, a 2013 systematic review concluded that only nitroimidazole drugs and the thiazolide drug nitazoxanide are likely to be beneficial to patients [10]. This fact, when coupled with emerging drug resistance to metronidazole as well as its lack of activity against the infectious cyst form of *E. histolytica* necessitates the search for new treatment options [13].

In this study we tested the hypothesis that *E. histolytica* could be killed by FDA-approved antineoplastic kinase inhibitors, possibly via action on parasitic homologs of human kinases. Out of 24 such drugs tested, six were shown to possess strong anti-amoebic properties, representing a completely new class of drugs in this area. All of the six highly active drugs displayed unique and important advantages over the current treatment. Dasatinib, ibrutinib, bosutinib, ponatinib, neratinib, and olmutinib were all for the first time shown to kill *E. histolytica* trophozoites *in vitro* significantly more potently than metronidazole. Ponatinib, neratinib, and olmutinib in particular demonstrated sub-micromolar EC$_{50}$ values rarely observed for any compound against this organism. These latter three were also shown to act as rapidly as metronidazole, and two of them, ponatinib and neratinib, were shown to act after shorter exposure times. Significantly, ibrutinib was also shown to kill the cysts of the related model organism *E. invadens* in contrast to metronidazole which was not. This feature is particularly unique and desirable for epidemiological purposes. Outside of the current study, both ibrutinib and neratinib have been shown to possess good brain penetrance, as does metronidazole [26–28]. Taken together all these properties give the six drugs strong potential for repurposing against *E. histolytica* infection, especially in advanced amoebiasis cases where infection has progressed to the liver or brain, or in cases of cancer patients with *E. histolytica* infections [3,8,29].

Interestingly, a recent high-throughput screen of the reframeDB commercial drug library also observed activity of ponatinib and dasatinib against *E. histolytica*, as well as the tyrosine kinase inhibitor rebastinib not included in the current study. While this screen is currently unpublished, the results can be viewed at (https://reframedb.org/assays/A00203). The

researchers involved in this screen have also conducted screens of the same library against the parasitic amoebae *Naegleria fowleri* and *Balamuthia mandrillaris*, both of which found ponatinib to be active [30]. All of these results further validate the potential for AKIs as a new class of highly potent drugs against *E. histolytica*, and potentially other parasitic amoeba as well.

One drawback to AKIs which might limit their use as hypothetical clinical antiparasitic drugs is their ability to cause moderate adverse effects in humans. All of the drugs found to be highly active in this study are known to possess this downside [31–40]. This has been observed both in studies of cancer patients and of healthy volunteers, in which these drugs are nevertheless generally described as "safe and well-tolerated" [41–44]. In particular, AKIs tend to cause diarrhea as one of the most common adverse effects and as such have the potential to exacerbate the symptoms of a diarrheal disease such as amoebiasis [37–39]. This could possibly be due to disruption of the intestinal microbiome by the AKI drugs, which has been previously reported [45,46]. It is worth noting however, that this feature is shared in common with the current standard of care for amoebiasis, metronidazole [7]. As such it may not necessarily disqualify AKIs from use against this disease, especially as it has been found to be easily manageable in clinical trials with standard anti-diarrheal therapy [38]. Another strategy to circumvent the adverse effects associated with these AKIs could involve the testing of structurally related molecules for activity against amoebae without activity against human kinases.

Taken together the results of this study document a new class of FDA-approved drugs with strong potential for repurposing against a widespread and devastating pathogen. Future research may expand on these findings by characterizing the molecular mechanisms underlying the actions of these drugs as well as testing their *in vivo* efficacy.

## Supporting information

**S1 Fig. False-positive assay against *E. histolytica* trophozoites.** All drugs were tested at a serially-diluted range of concentrations. Cell viability measured at T = 0.
(TIF)

**S1 Dataset. Fig 1 Data.**
(XLSX)

**S2 Dataset. Fig 6 Data.**
(XLSX)

**S3 Dataset. Fig 7 Data.**
(XLSX)

**S4 Dataset. Fig 8 Data.**
(XLSX)

**S5 Dataset. Fig 9 Data.**
(XLSX)

**S6 Dataset. S1 Fig Data.**
(XLSX)

## Acknowledgments

The authors of this study would like to thank Dr. Jim McKerrow and the Center for Discovery and Innovation in Parasitic Diseases at the Skaggs School of Pharmacy at the University of California—San Diego as well as Lily Hahn, Thi Nguyen, and Abdolhakim Mohammed for their contributions to this work.

## Author Contributions

**Conceptualization:** Conall Sauvey, Ruben Abagyan.

**Data curation:** Conall Sauvey.

**Formal analysis:** Conall Sauvey, Da Shi.

**Funding acquisition:** Ruben Abagyan.

**Investigation:** Conall Sauvey, Gretchen Ehrenkaufer, Da Shi.

**Methodology:** Conall Sauvey, Gretchen Ehrenkaufer, Da Shi, Anjan Debnath.

**Project administration:** Conall Sauvey.

**Resources:** Ruben Abagyan.

**Software:** Da Shi.

**Supervision:** Anjan Debnath, Ruben Abagyan.

**Visualization:** Conall Sauvey.

**Writing – original draft:** Conall Sauvey.

**Writing – review & editing:** Gretchen Ehrenkaufer, Da Shi, Anjan Debnath, Ruben Abagyan.

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
