## [Decision Letter · Decision Letter 0]

14 Sep 2020

Dear Dr. Conall Sauvey,

 Thank you very much for submitting your manuscript "Antineoplastic kinase inhibitors: a new class of potent anti-amoebic compounds" for consideration at PLOS Neglected Tropical Diseases. As with all papers reviewed by the journal, your manuscript was reviewed by members of the editorial board and by several independent reviewers. In light of the reviews (below this email), we would like to invite the resubmission of a significantly-revised version that takes into account the reviewers' comments. 

We cannot make any decision about publication until we have seen the revised manuscript and your response to the reviewers' comments. Your revised manuscript is also likely to be sent to reviewers for further evaluation.

Sincerely,

Kiyoshi Kita

Associate Editor

Steven Singer

Deputy Editor

Reviewer's Responses to Questions

**Key Review Criteria Required for Acceptance?**

**Methods**

-Are the objectives of the study clearly articulated with a clear testable hypothesis stated?

-Is the study design appropriate to address the stated objectives?

-Is the population clearly described and appropriate for the hypothesis being tested?

-Is the sample size sufficient to ensure adequate power to address the hypothesis being tested?

-Were correct statistical analysis used to support conclusions?

-Are there concerns about ethical or regulatory requirements being met?

Reviewer #1: The manuscript describes about the screening of the FDA approves kinase inhibitors against the E. histolytica. The authors have used recently developed anti-cancer kinase inhibitors for this study, where they found most of these inhibitors were able to kill the parasite very efficiently. Very interesting way of finding new drugs for parasites.

Reviewer #2: The authors provide a clear and useful manuscript. They hypothesize that they can identify approved human kinase inhibitors that can be used as anti-amoebic compounds. They demonstrate that this is indeed the case with a number of kinase inhibitors. The experimental detail is sufficient and the experiments support their hypotheses and conclusions.

**Results**

-Does the analysis presented match the analysis plan?

-Are the results clearly and completely presented?

-Are the figures (Tables, Images) of sufficient quality for clarity?

Reviewer #1: The EC50 of the drugs was coming out in submicromolar ranges. The authors should use the concentration of the drug concentration near the EC50, why much higher concentration (10uM) was used for survival assays?

Why different concentration of drugs was used for different assays- Page 21 (line 352) and page 23 (line 406)

Reviewer #2: The results are clear and laid out well. The manuscript provides an accurate description of the work flow utilized and experimental validation of the questions at hand.

**Conclusions**

-Are the conclusions supported by the data presented?

-Are the limitations of analysis clearly described?

-Do the authors discuss how these data can be helpful to advance our understanding of the topic under study?

-Is public health relevance addressed?

Reviewer #1: (No Response)

Reviewer #2: Yes, the conclusions are supported by the new data presented in the manuscript. Potential limitations (side effects) of repurposing the best leads are described, and suggestions made for handling the limitations are presented. The work sets the stage for further experiments to look further amongst approved kinase inhibitors, look at potential dosing regimens and in vivo activity, look at non-approved (earlier stage) kinase inhibitors that may offer efficacy advantages, and identify the targets that are driving efficacy.

**Editorial and Data Presentation Modifications?**

Reviewer #1: Page16 (line 260-261) please mention reference and if there is not then what is the basis taken for the cutoff value chosen as 15?

 Fig. 8F shows that the effect of the drug is maximum at 36 hours, then why calculations were done with reference to effect of the drug at 48 hours?

 Please mention time for calculating EC50 values in figure legends for fig. 1 and 6

 Figure 6 is showing that there is almost 100% trophozoite inhibition when drugs are added but in figure 7 100% inhibition is not shown. please mention the reason for that.

 Resolution for Figure 5 and 8 is very bad, nothing is clear.

 I was not able to access supplementary fig 1.

 These are all AKI drugs, what can be the effect of these drugs when given to non cancer patients..

 As these drugs target kinases that have homologs present in bacteria, amoeba and animals what can be the effect of these drugs on intestinal microbes?

Reviewer #2: everything looks good

**Summary and General Comments**

Reviewer #1: Interestingly, most of these inhibitors are tyrosine kinase inhibitors. And amoeba genome has 372 kinases and about 250 phosphatases. There are several studies, any disturbance in the kinase-phosphatase signalling pathway could be fatal for the organism (Ahmad et al., 2020, Mansuri et al., 2014, 2016). Then why only few inhibitors are effective and others are not? Can the authors predict or perform, which kinases are inhibited? Which family of kinases are more crucial?

Reviewer #2: Thank you for this submission. I enjoyed reading your plan and the details of the experiments. You have set the stage for further work that could indeed lead to approval of a kinase inhibitor for this disease. These results should encourage others to dig into this more deeply and combine target and phenotypic based drug discovery in this context.

PLOS authors have the option to publish the peer review history of their article (what does this mean?). If published, this will include your full peer review and any attached files.

Reviewer #1: No

Reviewer #2: No
---

## [Decision Letter · Decision Letter 1]

21 Dec 2020

Dear Dr. Conall Sauvey,

We are pleased to inform you that your manuscript 'Antineoplastic kinase inhibitors: a new class of potent anti-amoebic compounds' has been provisionally accepted for publication in PLOS Neglected Tropical Diseases.

Best regards,

Kiyoshi Kita

Associate Editor

Steven Singer

Deputy Editor

Reviewer's Responses to Questions

**Key Review Criteria Required for Acceptance?**

**Methods**

-Are the objectives of the study clearly articulated with a clear testable hypothesis stated?

-Is the study design appropriate to address the stated objectives?

-Is the population clearly described and appropriate for the hypothesis being tested?

-Is the sample size sufficient to ensure adequate power to address the hypothesis being tested?

-Were correct statistical analysis used to support conclusions?

-Are there concerns about ethical or regulatory requirements being met?

Reviewer #1: The revised manuscript addressed all our questions and queries.

Reviewer #2: Yes, as I mentioned in my previous review

**Results**

-Does the analysis presented match the analysis plan?

-Are the results clearly and completely presented?

-Are the figures (Tables, Images) of sufficient quality for clarity?

Reviewer #1: (No Response)

Reviewer #2: Yes, ready to go

**Conclusions**

-Are the conclusions supported by the data presented?

-Are the limitations of analysis clearly described?

-Do the authors discuss how these data can be helpful to advance our understanding of the topic under study?

-Is public health relevance addressed?

Reviewer #1: (No Response)

Reviewer #2: Yes eady to go as my previous review

**Editorial and Data Presentation Modifications?**

Reviewer #1: (No Response)

Reviewer #2: as before

**Summary and General Comments**

Reviewer #1: (No Response)

Reviewer #2: looks good to go

PLOS authors have the option to publish the peer review history of their article (what does this mean?). If published, this will include your full peer review and any attached files.

Reviewer #1: No

Reviewer #2: No

---

## [Editor Report · Acceptance letter]

2 Feb 2021

Dear Mr. Sauvey,

We are delighted to inform you that your manuscript, "Antineoplastic kinase inhibitors: a new class of potent anti-amoebic compounds," has been formally accepted for publication in PLOS Neglected Tropical Diseases.

Best regards,

Shaden Kamhawi

co-Editor-in-Chief

Paul Brindley

co-Editor-in-Chief
